# You say tomato, I say the same: A large-scale study of linguistic accommodation in online communities

**Aleksandrs Berdicevskis**
Språkbanken Text
Department of Swedish,
Multilingualism, Language Technology
Gothenburg University
`aleksandrs.berdicevskis@gu.se`

**Viktor Erbro**
Chalmers University of Technology
`erbro@student.chalmers.se`

## Abstract

An important assumption in sociolinguistics and cognitive psychology is that human beings adjust their language use to their interlocutors. Put simply, the more often people talk (or write) to each other, the more similar their speech becomes. Such accommodation has often been observed in small-scale observational studies and experiments, but large-scale longitudinal studies that systematically test whether the accommodation occurs are scarce. We use data from a very large Swedish online discussion forum to show that linguistic production of the users who write in the same subforum does usually become more similar over time. Moreover, the results suggest that this trend tends to be stronger for those pairs of users who actively interact than for those pairs who do not interact. Our data thus support the accommodation hypothesis.

## 1 Introduction

Language is a tool not only for conveying information, but also for expressing attitudes, constructing identities and building relationships (Eckert, 2012). One manifestation of this fundamental property of language is that how we speak (or write) depends on whom we are speaking (or writing) to. How exactly the audience affects the linguistic production is a complex and multi-faceted process which can be approached from various perspectives. Consider, for instance, the audience design theory (Bell, 1984), social identity theory (Reid and Giles, 2008) and accommodation theory (Giles, 1973; Gallois et al., 1995).

In this paper, we perform a large-scale test of the hypothesis that people adjust their production style to their interlocutors. This phenomenon is known as *accommodation* (sometimes *attunement* or *linguistic alignment*) or *convergence* if the styles of the interlocutors are becoming more similar (*divergence* if they are becoming more different). While

it has received considerable attention within sociolinguistics (Rickford et al., 1994; Cukor-Avila and Bailey, 2001) and cognitive psychology (Garrod et al., 2018), large-scale longitudinal studies are wanting. An exception is a study by Nardy et al. (2014), who have observed a group of French-speaking children at a kindergarten for one year and shown that children who interacted more frequently adopted similar usages of a number of sociolinguistic variables (such as, for instance, the dropping of the consonant /R/ in post-consonantal word-final positions).

Internet and social media in particular provide us with a vast amount of data about how people communicate and how they use language for other purposes than information transmission (Nguyen and P. Rosé, 2011). While in some respects these data are not as informative as those collected by direct observation or experimenting, in some other respects they may be equally or even more useful, providing very detailed information about who interacted when with whom and how. Besides, it is often possible to collect large datasets that enable more systematic hypothesis testing.

We use data from a very large Swedish discussion forum (Flashback) to test a widely held sociolinguistic assumption that "the more often people talk to each other, the more similar their speech will be" (Labov, 2001, p.288). In brief, we find pairs of Flashback users which during some period of time have actively interacted (see Section 2.2 for the definition of "active interaction"). We define a measure of linguistic distance between users and show that it is valid for our purposes (see Section 2.3). For every pair of users, we then calculate the linguistic distance between the two users' production before they have started interacting ($\Delta_{before}$) and after it ($\Delta_{after}$), and the difference between these distances ($\Delta_i = \Delta_{before} - \Delta_{after}$). If the convergence assumption is correct, we expect that the distance will tend to become smaller and the aver-

age $\Delta_i$ will be positive.

A positive $\Delta_i$, however, can arise for different reasons, of which arguably the most prominent one is that distances between users become smaller not because users accommodate to specific interlocutors, but rather converge on a certain style adopted in the community (Danescu-Niculescu-Mizil et al., 2013). To test whether this is a better explanation, we perform a similar calculation for those pairs who have never had a single interaction, comparing texts written earlier ($\Delta_{early}$) and later ($\Delta_{later}$) during their activity on the forum ($\Delta_n = \Delta_{early} - \Delta_{later}$). If there is a convergence to norm, the average $\Delta_n$ should be positive.

It is also possible that both pairwise accommodation and convergence to the community norm occur simultaneously. Moreover, they might even be parts of the same process: if speakers do converge on a certain norm, this convergence can emerge (at least partly) due to pairwise interactions. It is, however, also possible that only one of these processes occurs. Speakers can, for instance, converge on the community norm by adjusting to some perceived "average" style and not specific individual interlocutors. On the other hand, it can be imagined that speakers do adjust to the individual interlocutors, but that does not lead to the emergence of the community norm (for instance, because different interlocutors are "pulling" in different directions). The purpose of this study is to provide some insight into these not entirely understood processes.

We envisage four likely outcomes of our experiments, summarized in Table 1. Other outcomes are possible, but would be more difficult to explain. We would, for instance, be surprised if $\Delta_n$ turns out to be larger than $\Delta_i$ (since if there is convergence to community norm, it should be affecting actively interacting and non-interacting users in approximately the same way). Another unexpected result would be a negative value of either $\Delta_n$ or $\Delta_i$, since that would imply systematic divergence (see discussion in Section 4).

## 2 Materials and methods

### 2.1 Corpora

We use Flashback,[1] a very large Swedish discussion forum covering a broad variety of topics which has existed for more than two decades. In 2021, the proportion of internet users in Sweden (excluding those younger than eight years) who visited the

forum at least once during the last 12 months was estimated to be 24% (Internetstiftelsen, 2021).

The forum is divided into 16 subforums, of which we use five in the main experiment: *Dator och IT* 'Computer and IT', *Droger* 'Drugs', *Hem, bostad och familj* 'Home, house and family', *Kultur & Media* 'Culture and media', *Sport och träning* 'Sport and training'. These five were selected as being relatively large, of comparable size and representing diverse and not directly related topics. In addition, we use a smaller subforum *Fordon och trafik* 'Vehicles and traffic' to evaluate our distance metric (see section 2.3).

To access the Flashback texts, we use the corpora created and maintained by Språkbanken Text, a Swedish national NLP infrastructure. The corpora are available for download[2] and for searching via the Korp interface (Borin et al., 2012) and its API.[3]

The basic corpus statistics are summarized in Table 2. The earliest available posts date back to 2000, and the corpora were last updated in February 2022. The number of users is estimated as a number of unique non-empty usernames. We list separately the number of "prolific" users, and we consider users prolific if they have written 6000 tokens or more. All other users will be discarded (many of the prolific users will not pass additional thresholds either, see Section 2.4).

Subforums may be further divided into subsub- and subsubsubforums, which we do not take into account. What is important for our purposes is that messages (posts) are always organized in threads: there is an initial message which starts a thread (often a question) and then an unlimited number of messages which either respond to the original message or to later messages or in some other way are related to the thread's topic. The structure of the thread is linear: that is, messages are posted in a strictly chronological order.

### 2.2 Defining interaction

Two users are assumed to have had an interaction if they have written messages within the same thread, the two messages are separated by no more than two other messages and there has gone no more than five days between the two messages were posted. This definition has been used by Hamilton et al. (2017) and Del Tredici and Fernández (2018),

[1] https://www.flashback.org/

[2] https://spraakbanken.gu.se/resurser?s=flashback&language=All
[3] https://ws.spraakbanken.gu.se/docs/korp

| | Outcome | Interpretation |
|---|---|---|
| 1 | $\Delta_i > \Delta_n > 0$ | Both pairwise accommodation and overall convergence to community norm are detected |
| 2 | $\Delta_i = \Delta_n > 0$ | No pairwise accommodation; overall convergence to community norm is detected |
| 3 | $\Delta_i > \Delta_n = 0$ | Pairwise accommodation is detected; no convergence to community norm |
| 4 | $\Delta_i = \Delta_n = 0$ | No pairwise accommodation; no convergence to community norm |

Table 1: Four likely outcomes of the experiment. $\Delta_i$ is the change of linguistic distance between actively interacting users, $\Delta_n$ is the change of distance between non-interacting users.

| Subforum | tokens | users | prolific users |
|---|---|---|---|
| Computer | 316M | 187K | 9.3K |
| Drugs | 257M | 123K | 8.0K |
| Culture | 434M | 211K | 12.2K |
| Home | 348M | 168K | 10.0K |
| Sport | 251M | 105K | 5.4K |

Table 2: Basic statistics about the Flashback subforums. Prolific users have written 6000 tokens or more

but without the temporal threshold. We consider the temporal threshold useful, since Flashback can have very long threads, sometimes spanning over the years.

See the definition of "actively interacting users" in section 2.4.

## 2.3 Measuring linguistic distance

**Potential solutions.** A traditional sociolinguistic approach would be to identify a number of linguistic variables (features for which variation is known to exist) and use them for comparison (Nardy et al., 2014).The main problem with this approach is that most variables are not very frequent and it is thus difficult to collect enough observations. A traditional NLP approach would be to use a language model (Danescu-Niculescu-Mizil et al., 2013). Here, the main problem would be to ensure that the model has enough training data. We use a metric which is often applied in authorship attribution studies, Cosine Delta (Smith and Aldridge, 2011), a modification of Burrows' delta (Burrows, 2002). Its main advantage is that it can often be successfully applied to relatively small datasets, and it is also computationally efficient. It can also be considered a language model, though a very simple unigram-based one.

**Cosine Delta.** To calculate Cosine Delta between two texts, the texts are represented as $t$-dimensional vectors where every element is a $z$-score (standard score) of the relative frequency of

one of $t$ most frequent words. The cosine of the angle between the two vectors gauges their proximity, by subtracting it from 1, we get the distance (see Equation 1).

$$\Delta_\angle(T, T') = 1 - \frac{\mathbf{z}(T) \cdot \mathbf{z}(T')}{||\mathbf{z}(T)||_2 ||\mathbf{z}(T')||_2} \quad (1)$$

Cosine Delta has been shown to outperform Burrows' Delta and other similar measures (Jannidis et al., 2015; Evert et al., 2015).

**Evaluating the metric.** A typical usage of Cosine Delta is to compare text X of unknown or disputed authorship with texts by authors A and B in order to see whose style is more similar to the one used in X and whether the similarity is strong enough to attribute the text. This is not the same task that we have in mind. We want to compare texts written by authors A and B at time P and then at a later time Q in order to see whether the styles of the two authors have become more similar. In other words, we are not trying to infer who authored which text (we know that). Instead, we want to be able to measure the distance between two different authors.

To test whether Cosine Delta is suitable for that, we run the following experiment. The main requirement for an evaluation is a meaningful benchmark which can represent the ground truth. In order to evaluate a distance measure we need a set of texts between which true distances are known. We create such a set by mixing texts produced by two authors in different proportions. For two Flashback users (A0 and A1), an equal amount of tokens is extracted and used to create six texts: Base (contains solely the A0 production), 1 (80% of production belongs to A0, 20% to A1; every token is randomly selected), 2 (60% A0, 40% A1), 3 (40% A0, 60% A1), 4 (20% A0, 80% A1) and 5 (100% A1), see Figure 1.

We accept as ground truth that the distance between the Base text and, say, Text 1 should be

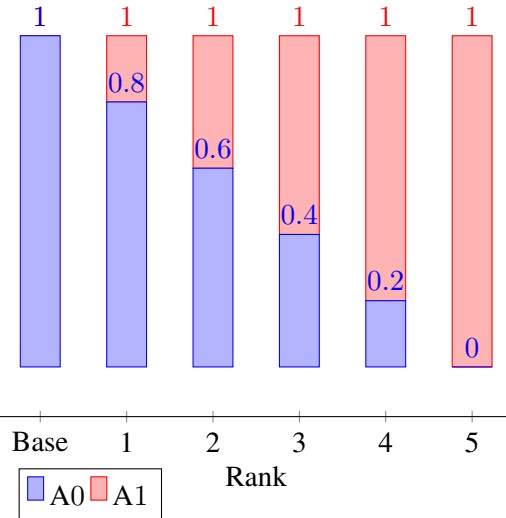

Figure 1: The artificial benchmark for evaluating the linguistic distance measure: six texts with different proportions of the authors' (A0 and A1) production.

| $n$ | $t$ | $\rho$ | $\Delta$ |
|---|---|---|---|
| 1500 | 150 | 0.936 (0.1) | 0.16 (0.06) |
| 1500 | 300 | 0.936 (0.1) | 0.15 (0.06) |
| 1500 | 450 | 0.940 (0.1) | 0.15 (0.06) |
| 1500 | 600 | 0.944 (0.1) | 0.15 (0.06) |
| 3000 | 150 | 0.950 (0.1) | 0.15 (0.07) |
| 3000 | 300 | 0.952 (0.1) | 0.14 (0.06) |
| 3000 | 450 | 0.952 (0.1) | 0.13 (0.06) |
| 3000 | 600 | 0.952 (0.1) | 0.13 (0.06) |
| 4500 | 150 | 0.976 (0) | 0.14 (0.08) |
| 4500 | 300 | 0.978 (0) | 0.13 (0.07) |
| 4500 | 450 | 0.978 (0) | 0.13 (0.07) |
| 4500 | 600 | 0.978 (0) | 0.13 (0.07) |
| 6000 | 150 | 0.994 (0) | 0.14 (0.06) |
| 6000 | 300 | 0.994 (0) | 0.13 (0.07) |
| 6000 | 450 | 0.994 (0) | 0.13 (0.07) |
| 6000 | 600 | 0.994 (0) | 0.13 (0.06) |

Table 3: Evaluating Cosine Delta on 50 ground-truth sets. $n$ is the number of tokens in the compared texts, $t$ is the number of frequent words used to construct the vector, $\rho$ is the average Spearman correlation coefficient, $\Delta$ is the average difference between authors A0 and A1 (between base and text 5). Interquartile ranges are provided in parentheses.

smaller than between Base and Text 5. We use Cosine Delta to compare Texts 1–5 with the Base text, rank them by their distance from Base and then measure Spearman correlation coefficient between this ranking and the true one (1, 2, 3, 4, 5).

We run the ranking test on 50 artificial sets, each consisting of six texts generated from two different authors' production, as described above. All data were extracted from the subforum *Fordon och trafik* 'Vehicles and traffic' (not used in the main experiment). The data were extracted consecutively without any randomization, i.e. the extraction script started from the beginning of the corpus, tried to extract a predefined number of tokens for every new user it encountered and stopped when it collected enough data for 100 unique users.

We try several combinations of two parameters: $t$, the dimension of vectors (the number of the most frequent words the frequencies of which will be used), and $n$, the minimum size of the texts to be compared (larger texts are expected yield more reliable estimates). The frequency list is compiled using the whole Flashback corpus (uncased). The results are reported in Table 3.

The performance of the ranking system is very high and increases as $n$ increases. Unfortunately, increasing $n$ decreases sample size, since less user pairs will be able to pass the thresholds (see Section 2.4). We judge that the best balance between reliability of Cosine Delta and sample size is reached with $n = 3000$ ($\rho \geq 0.95$). For $n = 6000$, the performance of Cosine Delta is better, but sam-

ple sizes (number of analyzable user pairs) are too small. We use $t = 300$, since larger values do not yield any gain for the chosen $n$ values. Using Pearson correlation coefficient instead of Spearman yields approximately the same results (the values are 1-2 percentage points lower, but the trends are almost the same).

We also calculate average distance between authors A0 and A1 (that is, between Base and Text 5) to obtain a very rough estimate of average distance between two different users. Later, when we measure how linguistic distance changes over time, we will use this estimate as a reference point, something to compare the change against, so that we can judge how large the effect size is. For $n = 3000$ and $t = 300$, the average distance is about 0.13 (though there is, unsurprisingly, considerable variation).

**Topic sensitivity**. An important potential problem with measures like Cosine Delta is that they are topic-sensitive, that is, the distance values can be affected not only by differences in the authors' styles, but also by the topic, i.e., what the specific texts are about (Mikros and Argir, 2007; Björklund and Zechner, 2017). This is extremely undesirable for

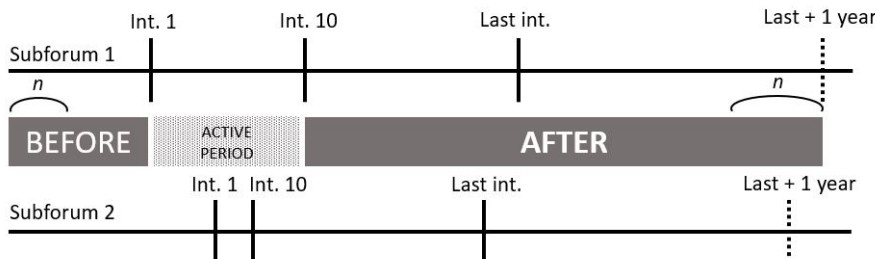

Figure 2: A visualization of how the periods **before** and **after** the active interaction has started are defined. Vertical lines represent interactions, the horizontal lines represent time. $n$ earliest tokens are sampled from the "before" period, $n$ latest tokens are sampled from "after"

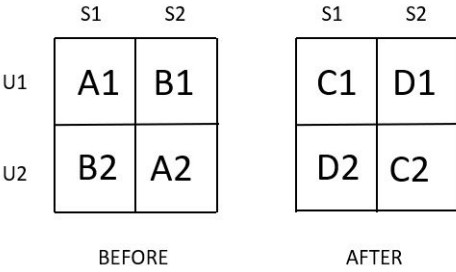

Figure 3: Visualization of the threshold requirements. Let the table cells represent how many tokens the **U**ser has written in the **S**ubforum in the given period. The following condition must be met for the user pair to be accepted: $((A1 \geq n\ AND\ A2 \geq n)\ OR\ (B1 \geq n\ AND\ B2 \geq n))\ AND\ ((C1 \geq n\ AND\ C2 \geq n)\ OR\ (D1 \geq n\ AND\ D2 \geq n))$

our purposes, since there is a risk that we observe that a convergence which is not in fact linguistic: the two authors do not start writing in a more similar way, they just start writing about more related topics. To eliminate or at least mitigate this risk, we always compare authors A and B by using texts that A wrote in one subforum and B in another subforum. While it is not completely impossible that the authors discuss similar topics in different subforums, it seems unlikely that "topical convergence" will systematically occur across subforums.

Note also that in the evaluation experiment described above all users come from the same subforum. Moreover, their production was extracted from the corpus consecutively and thus at least parts of it come from the same threads. That means that the users are likely to discuss related topics, and the ranking system must be able to capture differences in style despite potential similarities in topic, which it does very well.

## 2.4 Calculating distance change

As mentioned in Section 2.3, all our calculations are always based on two subforums at once (for instance, Home and Sport or Drugs and Computer). We will call such pairs of subforums *duplets* (to distinguish them from user *pairs*).

Two users are considered to have gone through a *period of active interaction* if they have had at least 10 interactions within a year in each of the subforums (that is, no less than 20 interactions in total). By requiring that the users actively interact in both subforums we ensure that there is a theoretical reason to expect convergence in both subforums and that the data are generally more comparable. We compare the production of users before and after the active interaction period, but ignore the period itself.

Within a subforum, the active period can have any length from one day to 365 days. We do not measure how often the users interact after the active period, but we discard all texts that have been produced more than one year later after the last interaction (it may be that users continue to interact and there are no messages to discard).

In other words, the general idea is that production before the active period includes everything written before the first interaction, production after the active period includes everything written after the tenth interaction (given that it is no more than one year apart from the first interaction), but no later than one year after the last interaction. We are, however, dealing with two subforums at once, and thus have two dates for each of the three seminal interactions. For convenience, we want the active period to be defined in the same way for both subforums. We achieve that by using the earlier of the dates for the first interaction and the later of the dates for the tenth generation (this can lead to

joint active period being longer than a year). When discarding the messages that were written after the users have stopped interacting (if any), we use the later of the last interaction dates. See the visual summary in Figure 2.

Users who have never had a single interaction are labelled as non-interacting. We compare them to actively interacting users and ignore all that end up in between: that is, have had some interactions but failed to pass the criteria outlined above (e.g. have had less than 10 interactions in total or have had more, but never 10 within a year). The reason for that is that we want the difference between groups (non-interacting and actively interacting users) to be as large as possible, so that potentially small effects can become visible.

Remember that we always want the linguistic distance to be calculated using text from different subforums. The procedure is as follows. For every pair, if before the active period, User 1 has produced at least $n$ ($n = 3000$) tokens in Subforum 1, and User 2 has produced at least $n$ tokens in Subforum 2, we calculate the distance between them, taking $n$ tokens for User 1 from Subforum 1 and $n$ tokens for User 2 from Subforum 2.

Obviously, if User 1 has $n$ or more tokens in Subforum 2, and User 2 has $n$ or more tokens in Subforum 1, the distance is calculated using tokens from Subforum 2 for User 1 and from Subforum 1 for User 2. If both conditions are met (Condition 1: User 1 has $n$ or more tokens in Subforum 1 and User 2 has $n$ or more tokens in Subforum 2; Condition 2: User 1 has $n$ or more tokens in Subforum 2 and User 2 has $n$ or more tokens in Subforum 1), we calculate both cross-subforum distances and use their arithmetic mean as the final result. If neither of the conditions is met, the pair is discarded. This procedure is visualized in Figure 3. The same user can occur unlimited times in different pairs.

Note that when we calculate distance between users A and B, we always use the same amount of tokens ($n$) for A and B (since using texts of different sizes might skew the Cosine Delta). For the "before" period, we extract the earliest $n$ tokens, for the "after" period, the latest $n$ ones (see Figure 2). The idea is to maximize the temporal distance between the periods in order to see stronger effect.

For non-interacting users, it is not obvious how to define "before" and "after", since the active period is not defined. We do the following: find the earliest first interaction date and the latest last in-teraction date across all actively interacting pairs. Then we take the date which is exactly in the middle between those two as the active period (the length of the active period is thus one day, which is common for interacting pairs, too). Then exactly the same procedure as for actively interacting pairs is applied, using the middle date to divide the data into "before" and "after".

There are many more non-interacting pairs than actively interacting ones, and calculating the distance change for all of them is computationally expensive. We go through the list of all non-interacting pairs in a randomized order and stop when $m$ pairs have met the conditions, where $m$ is five times the number of actively interacting pairs that have met the conditions. The reason for this decision is that the number of actively interacting pairs is rather small for some combinations of the subforums, and it makes sense to have somewhat larger samples at least for the non-interacting group.

## 3 Results

We perform the comparisons for all possible combinations of subforums (ten duplets in total). The results are summarized in Table 4. For every duplet and every type of user pair (actively interacting vs. non-interacting) we report sample size, average distance change ($\Delta_{before} - \Delta_{after}$) and the proportion of pairs for which the change was positive (the distance became smaller). Results for samples where the number of pairs is less than 20 are not reported.

Remember that in the evaluation experiment (Section 2.3) we roughly estimated the average distance between two different users to be around 0.13 for the chosen parameter values. While there clearly is large variation, and while the average distance can be larger for the main experiment (since the users' texts come from different subforums, not the same one), the estimate still provides us with a reference point and helps to put the observed distance changes in perspective. For Home-Sport-i, for instance, the average change is 0.033, which is approximately 25% of 0.13. This means that on average, actively interacting users in this duplet change their styles so much that they cover one quarter of an average distance the styles of two different persons.

Overall, the distance tends to become shorter both for interacting and non-interacting pairs. The

| Subforum1 | Subforum2 | type | pairs | positive | change | IQR |
|-----------|-----------|------|-------|----------|--------|-----|
| home | sport | i | 29 | 0.828 | 0.033 | 0.042 |
| home | sport | n | 145 | 0.524 | -0.012 | 0.081 |
| computer | drugs | i | 15 | - | - | - |
| computer | drugs | n | 75 | 0.680 | 0.048 | 0.096 |
| sport | drugs | i | 67 | 0.612 | 0.015 | 0.110 |
| sport | drugs | n | 335 | 0.546 | 0.002 | 0.094 |
| home | computer | i | 46 | 0.630 | 0.060 | 0.121 |
| home | computer | n | 230 | 0.617 | 0.029 | 0.089 |
| home | drugs | i | 22 | 0.682 | 0.101 | 0.201 |
| home | drugs | n | 110 | 0.664 | 0.028 | 0.081 |
| sport | computer | i | 89 | 0.607 | 0.031 | 0.153 |
| sport | computer | n | 445 | 0.600 | 0.027 | 0.105 |
| home | culture | i | 105 | 0.686 | 0.042 | 0.090 |
| home | culture | n | 525 | 0.608 | 0.020 | 0.078 |
| sport | culture | i | 332 | 0.506 | -0.014 | 0.119 |
| sport | culture | n | 1660 | 0.619 | 0.009 | 0.101 |
| drugs | culture | i | 25 | 0.680 | 0.077 | 0.190 |
| drugs | culture | n | 125 | 0.584 | 0.023 | 0.115 |
| computer | culture | i | 144 | 0.694 | 0.058 | 0.114 |
| computer | culture | n | 720 | 0.640 | 0.032 | 0.107 |

Table 4: Results across the subforum duplets. Listed: whether the pair of users actively **i**nteracts or **n**ot (**type**); total number of **pairs** in the sample; proportion of pairs for which $\Delta_{before} - \Delta_{after}$ is **positive**; average **change** $\Delta_{before} - (\Delta_{after})$ and the corresponding **IQR**. Shaded are rows where sample size is smaller than 20 pairs (considered unreliable)

| Subforum1 | Subforum2 | $\Delta_{pos}$ | $\Delta_{change}$ | Outcome | Comment | $p(\Delta_{pos})$ | $p(\Delta_{change})$ |
|-----------|-----------|----------------|-------------------|---------|---------|-------------------|----------------------|
| home | sport | 0.304 | 0.045 | 3 | div. for non-int.? | **0.002** | **0.013** |
| computer | drugs | - | - | - | sample too small | - | - |
| sport | drugs | 0.066 | 0.013 | 1 or 2 | | 0.180 | 0.177 |
| home | computer | 0.013 | 0.031 | 1 or 2 | | 0.485 | 0.134 |
| home | drugs | 0.018 | 0.073 | 1 | | 0.519 | **0.001** |
| sport | computer | 0.007 | 0.004 | 2 | small diff. | 0.491 | 0.409 |
| home | culture | 0.078 | 0.022 | 1 | | 0.076 | **0.013** |
| sport | culture | -0.113 | -0.023 | ? | div. for int.? | 1.000 | 0.994 |
| drugs | culture | 0.096 | 0.054 | 1 | | 0.242 | **0.018** |
| computer | culture | 0.054 | 0.026 | 1 | | 0.121 | **0.037** |

Table 5: Classification of outcomes (see Table 1) per duplet (see Table 4). $\Delta_{pos}$ = difference between the proportions of presumably accommodating pairs for interacting and non-interacting users (column **positive** in Table 4). $\Delta_{change}$ = difference between the average distance changes for interacting and non-interacting users (column **change** in Table 4). $p$-values are significance values obtained by bootstrapping (those below 0.05 are boldfaced). Positive values of $\Delta$s and small values of $p$s indicate Outcome 1.

proportion of pairs which (presumably) accommodate is larger than 0.5 in 19 cases out of 19 (though only marginally so for Sport-Culture-i). The average change is positive in 17 cases out of 19 (but note that IQR is very large in most cases, which means considerable variation across pairs).

We compare the observed results with the possible outcomes in Table 5. We concentrate on the effect size and the robustness of effect (how often the same pattern can be observed across duplets and thresholds) rather than statistical significance testing (see Wasserstein et al. (2019) about the limitations and pitfalls of this approach in general and Koplenig (2019) in corpus linguistics in particular). Nonetheless, we also calculate $p$-values to estimate how likely it is that the observed (or larger) differences between interacting and non-interacting pairs could have arisen by chance. We use a bootstrapping method: we randomly divide all pairs into two samples of the same sizes as the samples of interacting and non-interacting pairs 10,000 times and calculate the proportions of cases when $\Delta_{pos}$ and $\Delta_{change}$ are larger than or equal to actual values.

Out of nine duplets with sufficient sample size, seven demonstrate the effects which are compatible with either Outcome 1 (overall convergence to a community norm and pairwise accommodation on top of that) or Outcome 2 (just overall convergence) in Table 1. If we use the conventional 0.05 threshold for the $p$-values, then for four duplets (Home-Drugs, Home-Culture, Drugs-Culture, Computer-Culture) at least one of the two $p$-values is significant. We judge these four duplets to be most compatible with Outcome 1. In the Sport-Computer duplet, the differences are small, while $p$-values are large, which indicates no difference between interacting and non-interacting pairs, i.e. Outcome 2. For Sport-Drugs and Home-Computer, the differences are rather large, but the $p$-values are above the threshold, which makes it difficult to choose between Outcome 1 and Outcome 2. In the Home-Sport duplet, there is a clear difference, but the average distance change for non-interacting users is negative, suggesting divergence. The proportion of converging pairs is, however, marginally larger than 0.5. We label this case as Outcome 3: no clear effect for non-interacting users, thus no evidence for convergence to a community norm. Finally, the Sport-Culture duplet exhibits an unexpected effect: the non-interacting users seem to accommodate, while the interacting users do not (according to the proportion measures) or even diverge (according to the average change).

## 4 Discussion

From Section 3 it is clear that not all the results unambiguously point in the same direction. It is, however, obvious, that in most cases distance does become shorter, that is, users do converge. Negative results (distance becomes longer) are not only less frequent, but also weaker than most of the positive ones.

By comparing distance changes with the average distance between two different users we show that the effect sizes can be viewed as considerable.

The shortening trend tends to be stronger and more robust for actively interacting pairs, but in some cases there is not enough evidence to prefer Outcome 1 over Outcome 2.

More direct insight into the process of convergence would of course be desirable before it can be stated with certainty that it is *caused* by interactions. Nonetheless, our results provide evidence that it actually *can be so*. In other words, we show that convergence can exist (a necessary condition is meant: distance changes are observed), but not that it definitely exists.

Note that while a reversed causal link can be suggested: users who have similar writing styles will interact more often, or "birds of a feather flock together" (McPherson et al., 2001), it can hardly explain our results on its own: why would users who write on the same subforum and especially those who interact become linguistically closer over time?

There are several reasons to why our results are not as clean as one might want them to be (apart from the obvious "random noise"). First, users in the pairs that we label as "non-interacting" can still interact in other Flashback subforums. Second, while we showed that Cosine Delta is a very good measure for linguistic distance, the definition of an interaction is more arbitrary. There is already a tradition of using the "post-nearby-in-the-same-thread" measure (Hamilton et al., 2017; Del Tredici and Fernández, 2018), but it has not really been evaluated. Overall, further exploration of the same (or similar) data is of course desirable. Different experimental designs, different thresholds, different measures would show how robust the observed effects are.

We find the following questions particularly appealing for future studies.

- If we compare accommodation across interacting pairs, will it be correlated with the number/intensity of interactions?

- What happens if we consider not only direct connections between users, but also indirect ones? If A interacts with B, B interacts with C, but A does not directly interact with C: do A and C become closer?

- What happens if A and C from the previous example are pulling the style of B into different directions?

- Why do we sometimes observe negative values that suggest divergence (the distance increases)? Danescu-Niculescu-Mizil et al. (2013) observe an increasing divergence between the community norm and the production of a user who is become less active in the community (and will eventually leave), but it is unclear whether this can explain our results.

- Is it possible to explain convergence and divergence better if we take into account the content of the users' posts and the relationship between users?

## 5 Conclusions

We show that writing styles of users who participate in the same subforums do become more similar over time and that this increase in similarity tends to be stronger for pairs of users who actively interact (compared to those who do not interact), though this is not an exceptionless trend. These results support the accommodation hypothesis (let us repeat Labov's wording: "the more often people talk to each other, the more similar their speech will be").

It is desirable to see if the observed effects can be replicated in similar studies with different experimental settings.

All data and scripts necessary to reproduce the study are openly available.[4]

---

[4] https://github.com/
AleksandrsBerdicevskis/
LinguisticConvergence

## Acknowledgements

This work has been supported by the Cassandra project (funded by Marcus and Amalia Wallenberg Foundation, donation letter 2020.0060) and supported by the Swedish national research infrastructure Nationella språkbanken, funded jointly by the Swedish Research Council (2018–2024, contract 2017-00626) and the 10 participating partner institutions.

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
