# OpenReview forum: "You say tomato, I say the same: A large-scale study of linguistic accommodation in online communities"
_NoDaLiDa/2023/Conference — NoDaLiDa 2023_

### Official Review · Reviewer_jF5U · 2023-02-20
**Interesting paper on corpus-based testing of the "convergence hypothesis", stating that interacting speakers will tend to have more and more resembling language.**

**Rating:** 6
**Confidence:** 3

**Review:**

In this paper, the authors test the convergence hypothesis, a widely assumed sociolinguistic hypothesis that the more people interact, the more similar their way of talking will be. The test is done via a large Swedish corpus from a discussion forum.

The method (i) identifies pairs of users who have interacted within the forum for a certain period of time, (ii) defines a measure of linguistic distance, and (iii) compares the linguistic distance before and after interaction. A distance decrease will be in agreement with the convergence hypothesis.
To control whether this decrease is not a general convergence of the whole community, which would be independent from interpersonal interactions, the authors also measure the change in language distance over time, for the whole community, by using pairs of users who have not interacted.
This allows them to compare a measure of community convergence versus convergence after inter-personal interaction.

An interaction is assumed between two users if they have posted messages within the same thread, with thresholds for the number of intervening messages, and time span (2 messages and 5 days).

The chosen language distance is Cosine Delta, a measure borrowed from authorship attribution studies. Language is represented through a vector comprising the relative frequency of each of the t most frequent words. And these vectors are simply compared using cosine.
The authors stress that this similarity is obviously sensitive to the texts' topic, and not only to the linguistic characteristics.
To mitigate for this effect, texts of users in a pair are chosen from different topical subforums.

User pairs are drawn from 10 subforum pairs, and results are averaged within each subforum pair, both for interacting user pairs and non-interacting user pairs.

The authors then study how many subforum pairs show results that are compliant with the convergence hypothesis.


**Strengths**

- I found the proposed study very interesting

- Precisely described and well thought-out methodology

**Suggestions for improvement**

Some choices could be better motivated.
In particular the cosine delta metric, essentially a BOW technique, is only considering a lexical view on language convergence (even though the t most frequent words are likely to contain first the functional words).
This should be better emphasized in the paper.
Why not exclude content words altogether?

The time span thresholds should also be better motivated, possibly with links to the sociolinguistic litterature.

The alternative to computing statistical significance (lines 444-447) did not convince me. I still wonder whether randomly dividing user pairs between interacting and non-interacting pairs would result in different results.

The terminology is sometimes confusing. You measure language distance using cosine similarity.
I think that the delta defined in equation (1) should be the distance (namely 1 minus that, cf. line 207-208).
More generally I don't see the point in turning these into distances, why not use "sim_before" instead of delta_before?

The paper is unclear concerning previous work on corpus-based evidence of the convergence hypothesis. The introduction mentions there are "few". Few or none?

Couldn't it be possible to identify examples of accomadation, using e.g. the most accomodating user pairs?
Also what are the words in the vector space that account for most change?


**More detailed comments and questions:**


- paragraph 250-260: maybe it would have been better to ensure that the chosen users had not directly interacted?

- lines 243-249: on top of Spearman, Pearson's correlation would be useful to judge the metric, since the metric is going to be used in an absolute manner (no normalization). The Spearman's correlation seems to correspond to a weaker evaluation.

- lines 310-318: why didn't you simply use the same protocol for the evaluation of the metric and for the core experiments? I think it would be sounder.

- line 327: what is the intuition of imposing interaction in each of the subforums?

How did you choose the time span of 1 year? Does it have any link with previous work on the convergence hypothesis? It seems quite long a time span.

- line 407-408: is it really one **day**? then the maximum span of one year seems disproportionated.

- lines 443-444: "In addition, p-values will be affected by varying sample sizes": but your results too! this is the point of significance tests.
I think it would be better to provide the significance tests, even if inconclusive, and even if you have arguments against them.
The way it is formulated here, it seems you have interest to hide such results.



**Paper Type:**

Long paper

---

### Official Review · Reviewer_Hikr · 2023-02-27
**Interesting paper on how users style in social media converges over time.**

**Rating:** 8
**Confidence:** 5

**Review:**

The paper presents a very interesting analysis of the way users accommodate their style to each other and to a common norm over time. The method seems very interesting and novel. It applies a measure, Cosine Delta, that has been shown to be useful in authorship attribution, to the task of measuring the similarity between styles of interacting users, and how this similarity changes over time.
I have a few questions/comments
1) Where in the formula for Cos Delta is the cosine subtracted from 1 (as stated on line 207)?
2) What exactly is a token in all stats and calculations? Is it a word token, including presumably punctuation? If that is correct, it is maybe worth mentioning the fact that style is exclusively modeled in terms of word unigrams (other features of style are often used, such as sentence length, capitalization, lexical diversity, etc)
3) I don’t understand exactly in what way the procedure for selecting interacting pairs is applied to non-interacting ones (lines 409-410).
4) “non-interacting users do not” on line 489 should probably read “interacting users do not”.



**Paper Type:**

Long paper

---

### Official Review · Reviewer_Rh3B · 2023-03-10
**Exploring the validity of a socieloinguistics assumption in a  Swedish discussion forum.**

**Rating:** 7
**Confidence:** 3

**Review:**

This paper explores the assumption in sociolinguistics and cognitive psychology that the more people talk together, the more similar their communication becomes. To this end, they use a dataset from a large Swedish discussion forum, from which they select specific themes, and follow specific threads. They select the data in such a way to allow them to be able to compare persons' A and B types of writing in time t, and compare the same persons' A and B writing in time t+n, to see if their communication overtime has made them similar. They predefined four likely outcomes of their experiments, where either or both pairwise accommodation between users happen, and either or both convergence to the community norm takes place.
To compute the convergence or divergence of writing style between users, the authors rely on the Cosine Delta metric, that is usually used for authorship attribution.
They show that while the approach might not be bulletproof, it is quite clear that in most cases distances between pairwise interlocutors gets shorter over time, i.e. users of the platform do converge. The authors also discuss the negative results of the experiment, which were less frequent, and weaker. They conclude hat convergence between users can exist, but not definitely exist.

Pros:
- Interesting and well written paper. The authors have clearly stated they hypothesis and experiment setup, and gave a good discussion of the results and observations.

Cons:
- I do not see any.

**Paper Type:**

Long paper

---

### Decision · Program_Chairs · 2023-03-17

Accept